# Antiviral Therapy of COVID-19

**DOI:** 10.3390/ijms24108867

**Published:** 2023-05-16

**Authors:** Georgii Gudima, Ilya Kofiadi, Igor Shilovskiy, Dmitry Kudlay, Musa Khaitov

**Affiliations:** 1NRC Institute of Immunology, Federal Medico-Biological Agency, 115522 Moscow, Russiamr.khaitov@nrcii.ru (M.K.); 2Department of Immunology, N.I. Pirogov Russian National Research Medical University, Ministry of Health of the Russian Federation, 117997 Moscow, Russia; 3Department of Pharmacology, Institute of Pharmacy, I.M. Sechenov First Moscow State Medical University (Sechenov University), 119991 Moscow, Russia

**Keywords:** COVID-19, SARS-CoV-2, antiviral therapy, convalescent plasma, monoclonal antibodies, nucleoside analogs, fusion inhibitors, protease inhibitors, antisense oligonucleotides, siRNA

## Abstract

Since the beginning of the COVID-19 pandemic, the scientific community has focused on prophylactic vaccine development. In parallel, the experience of the pharmacotherapy of this disease has increased. Due to the declining protective capacity of vaccines against new strains, as well as increased knowledge about the structure and biology of the pathogen, control of the disease has shifted to the focus of antiviral drug development over the past year. Clinical data on safety and efficacy of antivirals acting at various stages of the virus life cycle has been published. In this review, we summarize mechanisms and clinical efficacy of antiviral therapy of COVID-19 with drugs based on plasma of convalescents, monoclonal antibodies, interferons, fusion inhibitors, nucleoside analogs, and protease inhibitors. The current status of the drugs described is also summarized in relation to the official clinical guidelines for the treatment of COVID-19. In addition, here we describe innovative drugs whose antiviral effect is provided by antisense oligonucleotides targeting the SARS-CoV-2 genome. Analysis of laboratory and clinical data suggests that current antivirals successfully combat broad spectra of emerging strains of SARS-CoV-2 providing reliable defense against COVID-19.

## 1. Introduction

The SARS-CoV-2-infection/COVID-19 pandemic continues to pose a threat to public health. The number of registered cases of infection exceeded 641 million, and more than 6.8 million infected people died. About 270,000 cases of infection are registered daily [1,2]. The main directions of COVID-19 counteraction are vaccination (prevention of the development of the disease) and treatment of infected patients. Knowledge of SARS-CoV-2 routes of transmission, its molecular organization, life cycle features, and cellular infection mechanisms allowed us to improve treatment efficacy using various types of drugs. These include convalescent plasma, monoclonal antibodies, antiviral drugs (fusion inhibitors, protease inhibitors, nucleoside analogues), interferons. The accumulation of practical experience and the results of scientific research made it possible to improve therapeutic strategies and provided the basis for the development of new type of antiviral drugs specifically blocking viral replication by suppressing the activity of the corresponding sites of the virus genome. The review examines different types of antiviral drugs used in the treatment of patients with COVID-19, their targets, as well as the development and application of promising specific antiviral drugs based on antisense oligonucleotides.

## 2. SARS-CoV-2 Life Cycle

Coronaviruses (CoVs) are single-stranded positive-sense RNA viruses known for their tremendous mutation potential and capability to cross the species barrier [3]. They belong to the order *Nidovirales*, suborder *Coronavirineae* and the family *Coronaviridae*. It is a highly diverse family of enveloped viruses, which are further divided into four genera: *Alphacoronavirus*, *Betacoronavirus, deltacoronavirus*, and *gammacoronavirus* [4]. The first two genera mainly infect mammals, gammacoronaviruses primarily infect avian hosts, and deltacoronaviruses are characterized by the ability to infect both birds and mammals [5]. Coronavirus infections mainly result in respiratory and enteric diseases [6]. The most common symptoms include fever, cough, dyspnea, malaise, fatigue, and sputum/secretion. [7,8] There are also reports on neurological manifestations that occur quite frequently [9].

### 2.1. Binding to Target Cell and Fusion

The main targets for SARS-CoV-2 are bronchial epithelial cells, pneumocytes, and upper airway epithelial cells [10]. The initial stage of infection is the attachment of the virion to the target cell and subsequent fusion with it. The SARS-CoV-2 S-protein binds angiotensin-converting enzyme 2 (ACE2) via its RBD domain and, being cleaved by host proteases, namely, transmembrane serine protease 2 (TMPRSS2), allows the virus to enter the cell [11]. The fusion of virus and cell membranes results in the penetration of the virus genetic material into the cytoplasm (Figure 1) [12].

### 2.2. Primary Translation of Viral Genomic RNA

SARS-CoV-2 has a relatively large genome (about 30,000 nucleotides) coding at least 14 open reading frames (ORFs) [13]. The 5′and 3′-terminals contain non-translated regions that include conserved regulatory elements necessary for genomic RNA replication. The 5′-end contains CAP (7-methylguanosine) and the 3′-end contains the so-called poly(A)-tail, which are necessary to maintain the integrity of genomic RNA in the cell cytoplasm, protecting it from the action of intracellular RNAases [14]. There are 2 large open reading frames (ORF1a and ORF1b) spanning about 70% of the genome. They encode 16 nonstructural proteins (nsp1-16) that are necessary for viral genome replication and mRNA transcription. The remaining 30% of genomic RNA serves as a matrix for the transcription of mRNAs encoding structural proteins (S, E, N, M), which shelter the virus and facilitate its entry into the cell, and also encode so-called subgenomic mRNAs (sgRNA), from which accessory proteins are translated [12,15,16].

After penetration of the genomic RNA into the cytoplasm, it translates, resulting in the formation of 2 polypeptides pp1a and pp1ab from the ORF1a and ORF1b, which then processed to nonstructural proteins. The processing is performed by two proteases encoded in the virus genome (PL^pro^ and M^pro^). These nonstructural proteins in turn form the replication–transcription complex (RTC) required for subsequent viral replication and mRNA transcription (Figure 1) [12,15].

### 2.3. Replication of Genomic RNA and Transcription of mRNA

Replication of viral RNA is performed by the RTC complex, the main functional role in which is played by the proteins nsp12-RNA-dependent RNA polymerase, nsp8-primase, and nsp14-exonuclease [17]. Replication results in the formation of a full-length genomic RNA of negative polarity, which functions as a matrix for the synthesis of the sense RNA. The synthesized genomic RNA is then used for translation of nonstructural viral proteins and also participates in the assembly of mature virions (Figure 1). Subgenomic RNAs (sgRNAs) also transcribed by RTC by interrupting the RNA synthesis process in the transcriptional regulatory sequence (TRS) regions [12,15].

Between the ORFs encoding structural proteins, there are at least 8 ORFs encoding the so-called accessory proteins [18]. The RTC complex, in addition to replicating the virus genome, transcribes mRNA of structural proteins S, E, M, and N and sgRNAs encoding accessory proteins. sgRNAs are formed during the synthesis of genomic RNA replication due to the fact that the RTC complex interrupts the process of RNA synthesis in the TRS-element region [12,15].

### 2.4. Translation

Structural proteins (S, E, M, and N) and accessory proteins are translated from individual mRNA molecules; the former are necessary for the assembly of mature virions, while the exact role of the latter is not fully established. Accessory proteins are not involved in viral replication, but one assumption is that they modulate the antiviral response of the cell. For example, at least 3 accessory proteins encoded by ORF3b, ORF6, and ORF9b may have the function of interferon antagonists, and the accessory protein encoded by ORF8 can bind to the major histocompatibility complex and abolish its presentation capacity. All this leads to suppression of the cell’s protective response to infection (Figure 1) [19].

### 2.5. Virion Assembly and Budding

Next, structural proteins S, E, M, and N are synthesized in the ER, which, together with genomic RNA, drive the assembly of new virions that bud into the endoplasmic reticulum-Golgi intermediate compartment (ERGIC) and are then released into the extracellular space via trafficking in secretory vesicles (Figure 1) [12,15,20]. The cytopathogenicity of SARS-CoV-2 has been shown to be associated with profound morphological remodeling of the cell secretory apparatus, mitochondria, and peroxisomes. In particular, virus-induced Golgi fragmentation leads to the formation of viral replication organelles. These structures are thought to create a microenvironment favorable to RNA synthesis and protect viral RNA from degradation and sensing by pattern recognition receptors of the innate immune system. These double-membrane structures have been shown to create an intracellular network associated with the endoplasmic reticulum (ER) [21].

## 3. Etiotropic Therapy COVID-19: Official Recommendations and Medications

For antiviral therapy of COVID-19, the preparations of immunoglobulin and monoclonal antibodies, nucleoside analogs, protease inhibitors, fusion inhibitors, antisense nucleotides are allowed in different regimens [22,23]. The antiviral drugs discussed in this section are shown in Table 1. In addition to a brief description of the mechanism of action, it also provides data on the current official status of these drugs and references to official recommendations, where applicable.

### 3.1. Plasma of Convalescents and Preparations Based on Monoclonal Antibodies

One of the possible methods of treatment of COVID-19 patients is immunotherapy using plasma from people who have already had the disease (convalescents). Treatment with plasma derived from patients recovered from viral infections was first described during the 1918 flu pandemic, and it was one of the first drugs tried early in the COVID-19 pandemic. The antiviral effect of this plasma is realized due to the fact that it contains antibodies against the virus that can neutralize it and prevent infection. At least 7 clinical studies of the effectiveness of such plasma have been conducted with the participation of more than 12,000 patients. In the 1st report on 5 severely ill COVID-19 patients treated with convalescent plasma containing neutralizing antibodies, an improvement in the clinical condition of all participants was noted, determined by a combination of changes in body temperature, consistent assessment of organ failure, oxygen partial pressure/fraction of inhaled oxygen, viral load, serum antibody titer, routine biochemical parameters blood, acute respiratory distress syndrome (ARDS), as well as ventilation and extracorporeal membrane oxygenation before and after transfusion of convalescent plasma [37]. In a multicenter open randomized clinical trial involving 103 patients with severe COVID-19 in China, there was no statistically significant difference in the time to clinical improvement within 28 days in patients receiving convalescent plasma compared to standard treatment, which included symptomatic control and supportive care for COVID-19 patients based on the Chinese national COVID-19 treatment guidelines, with the use of antiviral medications, antibacterial medications, steroids, and Chinese herbal medicines (51.9% vs. 43.1%) [38]. However, the study was stopped early due to the slow recruitment of patients, which limited the possibility of identifying clinically significant differences. A systematic review of these studies did not reveal the clinical efficacy of plasma [39]; however, a positive clinical effect was achieved in a subgroup of immunocompromised patients [40]. The low efficiency of plasma is probably due to the fact that it contains antibodies not only to the RDB domain of the S-protein, but also to other less significant epitopes of the virus. In recent international guidelines, the use of convalescent plasma is not recommended for treatment of COVID-19 patients [23]. Alternative approaches under study include the use of hyperimmune globulin obtained from convalescent plasma and monoclonal antibodies specific to SARS-CoV-2 [41,42].

A more preferable strategy is the use of neutralizing monoclonal antibodies. In contrast to the blood plasma of convalescents, such antibodies specifically target the vital epitope of the RBD virus S-protein domains. The drugs bamlanivimab and etesevimab contain monoclonal antibodies to the SARS-CoV-2 S-protein and are used in combination with each other for the treatment of mild and moderate COVID-19 [24]. Another combination of monoclonal antibodies, casirivimab and imdevimab, has also been tested in clinical trials involving patients with COVID-19. It has been demonstrated that the use of this combination reduces mortality [25].

Alternative therapeutic strategies involve modulating the inflammatory response in patients with COVID-19. Monoclonal antibodies directed against key inflammatory mediators, such as interferon-γ, interleukin(IL)-1, IL-6, and C5a, are designed to suppress the inflammatory response after SARS-CoV-2 infection in order to prevent organ damage. The IL-6 inhibitors tocilizumab and sarilumab have been best studied, and more than a dozen randomized clinical trials have been conducted [26,27,43]. Tyrosine kinase inhibitors are being studied for their effectivity and cardiac safety in patients with COVID-19 [44,45].

### 3.2. Interferons

Interferons (IFNs) are signaling molecules, which represent an important component of innate immunity and provide first line defense against viruses. Three types of IFNs were described. Type I IFNs include IFN-α, IFN-β, IFN-ϵ, IFN-ω, and IFN-κ. Type II IFN family includes only one member—IFN-γ. IFN-λ1, IFN-λ2, IFN-λ3, and IFN-λ4 compose type III IFNs [46,47]. Mainly types I and III provide antiviral defense of the host [48,49]. Mammalian cells recognize the viruses through the pattern recognition receptors (PRRs) (like: TLRs, RLRs, and NLR), that lead to production of cytokines and IFNs [50]. IFNs could be produced by many cell types, but the main producers are considered macrophages and dendritic cells [48,49,51].

IFNs do not possess direct antiviral properties but realize an antiviral effect through different mechanisms. Viral infection induces the secretion of IFNs, which bind to the cell surface receptors and activate JAK/STAT signaling pathways. For example, binding of IFN-I to its receptor results in the activation of kinases JAK1 and TYK2, which phosphorylate factors STAT1 and STAT2, triggering their dimerization. These dimers then become capable of translocating into the nucleus, where they interact with regulatory elements on the chromosomes, activating the so-called interferon-stimulated genes (ISGs). ISGs encode antiviral proteins which limit viral invasion and restrict replication of the viral genome and viral protein translation [46]. Additionally, IFNs regulate antigen presentation, the function of natural killer cells, and help to activate B- and T-cells contributing to the viral clearance. Type III IFNs induce similar pathways, but they use other receptor chains [52].

Pivotal in vitro experiments showed that IFN-I inhibits SARS-CoV-2 replication [53]. Similarly, IFN-III suppressed SARS-CoV-2 replication in vitro [54]. Clinical observations also confirmed the contribution of types I and III IFNs to defense against SARS-CoV-2. Deficiency of I type IFNs in COVID-19 patients was associated with increased disease severity [55,56]. Poor IFN-I response to the infection could be the result of inborn genetic errors [57] or deficiency in plasmacytoid dendritic cells (important source of IFNs) [58]. Uncontrolled exploratory study showed that nebulized IFN-α2b in adult patients hospitalized with COVID-19 significantly reduced the duration of the detectable virus in the upper respiratory tract and duration of elevated levels of inflammatory markers (IL-6 and CRP) in the blood [59]. These initial findings suggested that IFNs could be used for COVID-19 therapy.

Subsequent controlled clinical trials have described in more detail its effectiveness for treating COVID-19. The published systematic review [60] summarizes 8 randomized controlled clinical trials of efficacy and safety of IFN-β in the treatment of hospitalized patients with moderate to severe COVID-19 [61,62,63,64,65,66,67,68]. A total of 4917 patients were included in these studies, 2490 of whom received IFN-β subcutaneously or inhaled. The performed analysis showed no beneficial effects of IFN-β in treatment of COVID-19 patients. The therapy did not improve the primary clinical outcome; the 28-day all-cause mortality rate of patients receiving IFN-β was similar to the control group [60]. Moreover, INF-β did not reduce the requirements of MV (mechanical ventilation) or ECMO (extracorporeal membrane oxygenation), did not increase the rate of survival to hospital discharge, and did not shorten the time to clinical improvement and length of hospital stay [60]. A detailed analysis showed no benefits of different routes of administration (subcutaneous injections or inhalations). However, the subgroup of patients with severe COVID-19 receiving INF-β therapy was associated with a lower mortality rate than the control group [60]. Additionally, INF-β decreased ICU (intensive care unit) admissions in hospitalized patients, but did not reduce the mortality [60], coinciding with a previous retrospective cohort study [69]. The performed safety analysis showed that INF-β therapy had a risk of adverse effects, which is similar to other therapies. The authors conclude that IFN-β is safe but does not appear to provide an increased survival benefit in hospitalized COVID-19 patients; however, it reduced the risk of ICU admission [60].

Another double-blind, placebo-controlled study investigating the safety and efficacy of type III IFNs was conducted. Mild-to-moderate COVID-19 patients received type III IFNs subcutaneously; they demonstrated a substantial decrease in the SARS-CoV-2 RNA level from day 3 after the beginning of the therapy. By day 7, the majority of participants (79%) treated with IFN-III had an undetectable viral load, compared with the placebo group (38%) [70]. These data show that type III IFNs have the potential to shorten the duration of viral shedding, but more clinical studies are needed to comprehensively assess efficacy of these types of interferons.

Summarizing the described above clinical data, we can conclude that the efficacy of type I IFNs is not obvious, while type III IFNs are needed to be investigated more intensively. One of the reasons for controversial clinical efficacy of IFNs could be the fact that viruses including SARS-CoV-2 possess the mechanisms allowing them to escape the antiviral effects of IFNs. SARS-CoV-2 can protect itself from PRR recognition that abolishes IFN induction. For example, viral genome replication occurs in membrane vesicles that interfere its binding to TLR3 [71]. Another mechanism exploited by SARS-CoV-2 is connected with the inhibition of IFN signaling by viral proteins (nsp12, ORF6, ORF8, and N) [72,73].

Despite the potential benefits of IFNs as antiviral agents, the flip side of the coin is that prolonged overexposure to IFNs increases pulmonary inflammation, leading to tissue damage. IFNs reduce epithelial cell proliferation and differentiation during viral infection, which makes the host more susceptible to bacterial invasions followed by lung injury [74]. Thereby, timing and duration are extremely important parameters which should be considered for IFN antiviral therapy implementation into clinical practice. Currently, WHO and ERS guidelines do not recommend the use of interferons for COVID-19 treatment [22,23]

### 3.3. Nucleoside Analogs (Inhibitors of Replication)

#### 3.3.1. Favipiravir

Favipiravir is a modified pyrazine analog that was initially approved for therapeutic use in influenza and Ebola [75,76]. The drug inhibits RNA-dependent RNA polymerase (RdRp), which is necessary for transcription and replication of viral genomes [77]. The antiviral activity of favipiravir in ARVI has been demonstrated previously [28]. With the wide spread of SARS-CoV-2, favipiravir was one of the first medications studied for the treatment of COVID-19. Q. Cai and colleagues were among the first to report on the experimental treatment of COVID-19 with favipiravir [78]. They found that antiviral therapy with favipiravir reduces the time of elimination of the virus. According to ClinicalTrials.gov (accessed on 18 April 2023), 34 clinical trials have been completed.

As a promising therapy, modified nucleoside analogues are used, which terminate the process of chain building by virus polymerase. However, this approach has its limitations, since the RNA polymerase of the virus has exonuclease activity to correct mutations. Favipiravir is not currently included in the latest international therapy guidelines for COVID-19 [23,79].

#### 3.3.2. Ribavirin

Ribavirin is a synthetic nucleoside analog of ribofuranose, which has activity against RNA-containing viruses, primarily hepatitis C virus. Ribavirin incorporates in viral RNA, and thereby inhibits its synthesis, causing mutations in the viral genome and suppressing normal replication of the virus.

Based on the docking analysis [29], it was proved that ribavirin can bind to the SARS-CoV-2 replication enzyme (RdRp), while the binding energy is comparable to that for native nucleotides from which the virus genome is synthesized. This means that ribavirin can inhibit the replication process of SARS-CoV-2.

However, one of the side effects of ribavirin is a decrease in hemoglobin concentration, which is unacceptable for patients with respiratory diseases [80]. Given this fact, as well as the limited effectiveness of ribavirin, revealed in 2003 during outbreaks of SARS, this drug is currently not used for the treatment of COVID-19 [23,81].

#### 3.3.3. Remdesivir

Remdesivir is an analogue of adenosine, incorporating in the genomic RNA chain of the virus during its replication, which eventually leads to the so-called “breakage” of the synthesized chain, and as a consequence, to disruption of the reproduction process of the virus.

Remdesivir was invented by Gilead Sciences to treat infections caused by Ebola and Marburg viruses. Later, its ability to suppress the replication of the SARS-CoV-2 clinical isolate was demonstrated in in vitro experiments in cell culture [82] and in vivo on a rhesus macaque infection model [83]. At the beginning of the pandemic of a new strain of coronavirus in the United States, one patient received remdesivir on the 11th day of illness due to the ineffectiveness of previous treatment. On the 12th day of infection, the patient’s clinical condition improved, while no adverse events were recorded [84]. Further, large-scale multicenter clinical trials of remdesivir were conducted with the participation of patients with COVID-19 [85]. The results revealed an improvement in the clinical condition in 36 of 53 patients (68%) treated with remdesivir [85]. It has also been shown that the late initiation of antiviral therapy with remdesivir is successful in the treatment of patients with severe COVID-19 [86]. However, a randomized, double-blind, placebo-controlled multicenter study conducted in ten clinics in China did not show statistically significant benefits of remdesivir in adult patients with confirmed SARS-CoV-2 infection [85]. The study of the drug is ongoing; according to ClinicalTrials.gov (accessed on 18 April 2023), 116 clinical trials of remdesivir are being conducted.. Remdesivir administration should be recommended within a week of onset of symptoms. Remdesivir may also be conditionally recommended for patients with severe COVID-19, and not recommended for patients with critical COVID-19 [23].

#### 3.3.4. Molnupiravir

Merck (MSD) and Ridgeback Biotherapeutics have developed the drug Molnupiravir, a broad-spectrum antiviral agent. This oral capsule preparation is a prodrug—a nucleoside analogue of β-D-N4-hydroxycytidine (NHC). Molnupiravir is able to introduce ‘errors’ into the genomic RNA of coronaviruses by replacing G nucleotides with A and C with U during replication. An increase in the frequency of mutations is associated with the antiviral effect of the drug. The study of the effect of NHC-5′-triphosphate (NHC-TP) on the activity of RNA-dependent RNA polymerase SARS-CoV-2 has generally confirmed this mechanism [87]. The efficacy of molnupiravir in clinical trials was 90% [88]. At the end of December 2021, the FDA granted permission for the emergency use of molnupiravir for the treatment of COVID-19 of mild to moderate severity in adult patients who are at high risk of disease progression to severe, including hospitalization or death, and for whom alternative permitted treatment options are unavailable or clinically unsuitable [89]. Treatment with molnupiravir is recommended as soon as possible after the diagnosis of COVID-19 and within 5 days of the onset of symptoms. FDA experts note that molnupiravir is not a substitute for vaccination. Molnupiravir is currently recommended for use in patients with non-severe COVID-19 at high risk of hospitalization [23].

### 3.4. Small Molecule-Based Therapy

#### 3.4.1. Protease Inhibitors

For many viruses, the protease enzymes play a critical role in viral protein maturation by cleaving translated polypeptides to functional products. As a result, viral proteases are often potential drug targets. For instance, HIV-1 protease inhibitors (tipranavir, darunavir, amprenavir, lopinavir, saquinavir, atazanavir, indinavir, ritonavir, and nelfinavir) [90] and hepatitis C virus (HCV) NS3/4A protease inhibitors (boceprevir, telaprevir, ritonavir, asunaprevir, paritaprevir, grazoprevir, glecaprevir, voxilaprevir, and sofobuvir) [91] are amongst the FDA-approved drugs. In the context of the pandemic, many researchers have turned their attention to existing viral enzyme inhibitors to evaluate their potential in COVID-19 therapy.

SARS-CoV-2 main protease M-pro (also called 3CL protease) catalyzing the cleavage of viral polyproteins into nonstructural proteins plays a crucial role in viral replication. High conservation of M-pro among SARS-CoV-2 strains and related viruses and the fact that this enzyme does not exist in humans makes it an attractive target for antiviral drug development [12]. In the beginning of the pandemic, the strategy of drug repurposing allowed one to establish M-pro inhibitors which were previously developed against SARS-CoV, MERS-CoV, HCV, and HIV-1 [92,93]. In the last 2 years, based on computational methods, thousands of compounds have been suggested as M-pro inhibitors [94]. More than 1700 compounds were experimentally validated and IC50 was established [95]. M-pro inhibitors could be divided into covalent and non-covalent types. The first ones covalently bound to catalytically important amino acids of the M-pro enzyme, resulting in its functional inactivation, while the second ones interact with M-pro through other bounds. The majority of inhibitors are non-covalent, but usually they exhibit lower activity compared to covalent ones [95].

SARS-CoV-2 M-pro represents 306 aa proteins consisting of three domains (I, II, and III). The catalytic center comprised of two important amino acids (His41 and Cys145) locates between domains I and II, while domain III provides dimerization, which is necessary for exhibiting of catalytic activity [96]. To reveal molecular mechanisms of M-pro inactivation, Günther and colleagues performed comprehensive co-crystallization experiments followed by obtaining x-ray diffraction datasets for more than 2000 compounds [97]. The majority of compounds with proven antiviral activity were found in the catalytic center of the enzyme, wherein some of them (especially peptidomimetics) were able covalently to bind to catalytically important Cys145. At the same time, another part of the biologically active compounds was bound outside the M-pro catalytic site. These data indicate that along with the catalytic center, other sites within M-pro could be critically important for its enzymatic activity, for example, domains involved in the maintaining of correct structure and dimerization of the enzyme. This explains why compounds binding outside the catalytic site exhibit antiviral activity. Authors established two allosteric binding sites of M-pro. The first is in the C-terminal dimerization domain and represents a hydrophobic pocket formed by Ile213, Leu253, Gln256, Val297, and Cys300. Compounds exploiting this site are inserting their aromatic moiety into this pocket that interfere with M-pro performance, as the integrity of this pocket is crucial for enzyme activity [98]. The second allosteric site represents the deep groove between catalytic and dimerization domains. Arg298 is one of the important amino acids of this site involved in the dimerization process. Compounds binding to this M-pro site interfere with the dimerization process and therefore, the catalytic activity of the enzyme [97].

##### Lopinavir/Ritonavir

Lopinavir is a protease inhibitor initially developed against HIV-1 and approved for AIDS treatment in combination with ritonavir, which prolongs the half-life period of lopinavir [30]. Lopinavir specifically inhibits the M-Pro of SARS-CoV-2, and inhibition of this enzyme is not associated with off-target effects due to the lack of known human analogues. Ritonavir is added to this combination to enhance the action of the main component through the inhibition of cytochrome P450 CYP3A4, which prolongs the half-life period of lopinavir, resulting in a more pronounced suppression of SARS-CoV-2 replication [31]. Lopinavir in combination with ritonavir has been proposed as a promising drug against COVID-19, especially after confirmation of anti-coronavirus activity in in vitro experiments [99].

In the beginning of the pandemic, lopinavir/ritonavir was applied for the treatment of COVID-19 and showed promising results [100]. In a clinical trial involving 47 patients with severe COVID-19 in China [101], the clinical effect in patients was accelerated elimination of the virus. Yet, further studies demonstrated the absence of clinical benefits [102,103].

Another randomized, controlled, open-label clinical trial involving 199 adult patients with severe COVID-19 showed no significant effect of treatment with the lopinavir/ritonavir combination [21]. A published systematic review [32] that summarized the results of 14 clinical studies included 1634 COVID-19 patients. Performed analysis revealed no clinical efficacy of lopinavir/ritonavir treatment compared to patients who received non-antiviral drugs in the following outcomes: negative rate of PCR, PCR negative conversion time, rate of improvement on the chest CT, rate of cough alleviation, disease progression, and hospital stay [32]. Additional analysis showed no differences in clinical outcomes between lopinavir/ritonavir versus chloroquine (of hydroxychloroquine) interventions [32]. However, therapy with lopinavir/ritonavir plus arbidol demonstrated significant benefits over lopinavir/ritonavir alone [104]. Other studies showed that the addition of lopinavir/ritonavir to other treatments could be beneficial. The combined therapy with lopinavir/ritonavir plus IFN-α shortens the duration of SARS-CoV-2 shedding [105]. Triple therapy with lopinavir/ritonavir plus IFN-b plus ribavirin showed superiority to lopinavir–ritonavir alone in alleviating symptoms and shortening the duration of viral shedding and hospital stay in patients with mild to moderate COVID-19 [62]. According to ClinicalTrials.gov, a significant number of studies on the efficacy of the lopinavir/ritonavir combination for COVID-19 therapy have been reported. Published data provided no sufficient evidence of lopinavir/ritonavir effectiveness in the treatment of COVID-19 patients. However, some benefits could be achieved if it is used in combination with other therapies. Current WHO guidelines do not recommend the use of lopinavir/ritonavir for the treatment of COVID-19 patients [23].

##### Nirmatrelvir/Ritonavir (Paxlovid)

Complete genome sequencing [106] and the revealing of the three-dimensional structure of SARS-CoV-2 M-pro [96] allowed the development of inhibitors with high specificity to this pathogen. One of the most active described M-pro inhibitors is the compound PF-00835231 representing peptidomimetic, initially developed against SARS-CoV-1. It suppresses SARS-CoV-2 M-pro as well in a covalent manner [107]. Clinical trials of lufotrelvir (which contains compound PF-07304814, which is a prodrug that metabolizes to PF-00835231) have already started [108,109], but there are no published data yet. The main obstacle for implementation of lufotrelvir into clinical practice is the intravenous administration route [110]. Therefore, another orally active specific inhibitor of the viral protease M-pro nirmatrelvir (PF-07321332) (Paxlovid) was developed [111]. Previously, there were similar drugs with an antiviral effect and using the same substrates for binding to proteins. The best known analogue, for which specific activity against SARS-CoV-2 has also been shown, is Boceprevir, which has passed clinical trials and is registered as an antiviral agent in the complex therapy of viral hepatitis C [112]. Thus, there are many years of experience in the use of inhibitors similar to Paxlovid.

Nirmatrelvir was approved for clinical use in combination with its pharmacokinetic booster ritonavir [111]. Clinical trials of nirmatrelvir/ritonavir were conducted in nonvaccinated COVID-19 patients with a high risk of progression to severe disease. These patients were enrolled to the study within 5 days of symptom onset. This therapy substantially reduced the proportion of hospitalized participants and mortality rate compared to a placebo group [33]. In another study, participants with low or standard risk of progression to severe disease were enrolled (some participants were vaccinated against COVID-19). The nirmatrelvir/ritonavir therapy showed no difference in the proportion of individuals achieving sustained alleviation of symptoms between treated and placebo groups [34].

It is important to note that anti-protease therapies have a higher barrier to resistance development than those observed with anti-spike monoclonal antibodies due to the low variability of M-pro. Emerged amino acid substitutions in the M-pro did not reduce nirmatrelvir activity in biochemical assay [34]. In vitro assays showed the activity of nirmatrelvir against the ancestral SARS-CoV-2 strain and the five variants of concern including Omicron [113]. Animal studies confirmed the antiviral activity of nirmatrelvir against Omicron lineages [114]. Conducted clinical trials revealed no significant associations between M-pro mutations and treatment efficacy [34].

Summarizing published data, we may conclude that lopinavir/ritonavir initially developed against other viruses are not effective in the treatment of COVID-19 patients, but could be beneficial when used in combination with other therapies. Nirmatrelvir/ritonavir designed against SARS-CoV-2 effectively reduce the mortality or hospitalization rate of COVID-19 patients, but more studies are needed for comprehensive analysis of their efficacy. Currently, WHO guideline recommends nirmatrelvir/ritonavir to treat COVID-19 as they may have greater efficacy in preventing hospitalization than the alternatives [23].

##### PLpro Inhibitors

Another SARS-CoV-2 protease—papain-like protease (PLpro)—is also involved in the cleaving of viral polyprotein during its life cycle. This fact makes PLpro an attractive drug target for antivirals as well [35]. Despite Mpro and PLpro being cysteine proteases, they are structurally different and cleave distinct sequences. Moreover, unlike Mpro, PLpro is able to process not only viral, but also host polypeptides [12]. PLpro enzymes from SARS-CoV and SARS-CoV-2 are very similar in their sequences and structures; therefore, PLpro inhibitors are able to suppress both strains with almost identical efficacy [35].

A recently published review summarizes the antiviral activity of the PLpro inhibitor compounds. This review does not discuss a large set of in silico studies based on docking analysis but focuses on studies which provide experimental confirmation of antiviral activity [35]. The majority of described PLpro inhibitors (tanshinone and its derivatives, asunaprevir, simeprivir and grazoprevir, famotidine, GRL-0617, etc.) reduced viral replication poorly, suggesting that use in clinical practice would be challenging.

Famotidine is the only PLpro inhibitor which reached clinical trials for the treatment of COVID-19 [35]. Interestingly, this compound was initially approved as the histamine H2 receptor antagonist for the treatment of other pathologies. A couple of clinical observations demonstrated that the use of famotidine was associated with clinical improvements in COVID-19 patients [115] and allowed the repurposing of this drug for COVID-19 treatment. However, the mechanisms of observed effects are undefined, as famotidine does not inhibit virus replication in vitro despite binding to PLpro [116]. Additionally, the results of clinical studies are yet to be published. Taking these into account, there are no reliable data demonstrating clinical efficacy of PLpro inhibitors (including famotidine) for COVID-19 treatment.

#### 3.4.2. Fusion Inhibitors (Umifenovir)

Umifenovir (Arbidol trademark) is an antiviral drug for the treatment of influenza infection approved in Russia and China. Chemically, it has an indole core, functionalized in all positions except one by different substituents. The drug inhibits the penetration of the virus into target cells and stimulates the immune response. Although a systemic meta-analysis of sixteen clinical trials of the efficacy and safety of umifenovir found no significant benefit for patients with COVID-19 [117], a recent phase III clinical trial showed that treatment with umifenovir is effective, safe, and well tolerated in patients with mild asymptomatic COVID-19 [118]. It is necessary to perform additional large-scale clinical trials for confirmation of the efficacy and safety of umifenovir in COVID-19 treatment.

## 4. Antisense Oligonucleotides

All the above-mentioned drugs or their analogs were originally created for the therapy of other viral infections, and their use for the treatment of COVID-19 was the result of repurposing their main biological effect, which most likely explains its limited efficacy against SARS-CoV-2. Decoding the SARS-CoV-2 genome and studying the biology of this virus in detail at the molecular level have opened up new opportunities in creating drugs specific to this pathogen. One of the most promising approaches for specific therapy of viral diseases is antisense technology based on the use of synthetic oligonucleotides, specifically targeting the viral genome. The idea of creating such preparations has been expressed by many specialists [119,120,121]. Some candidate drugs are under development or have even undergone preclinical studies [122,123]. However, results of clinical trials are currently available only for the drug MIR 19^®^ (siR-7-EM/KK-46) [124].

This is a novel antiviral drug, which targets the region of the SARS-CoV-2 genome encoding polymerase RdRp by the RNA interference mechanism (Figure 2). The RNA interference is a phenomenon of negative regulation of gene expression at the post-transcriptional level by small interfering RNAs (siRNA) [125]. siRNA (usually 20–25 bp in size) penetrating the cytoplasm complementary bind to the target mRNA followed by its exonuclease cleavage by the AGO protein of the multi-subunit RISC complex that results in mRNA degradation and blocking of the translation of viral proteins.

The antiviral formulation includes the synthetic siRNA against RdRp complexed with cell-penetrating cationic peptide, providing the cytoplasm delivery of the former. Preclinical studies showed that this complex substantially reduced SARS-CoV-2 replication in vitro (up to 10,000 times). In vivo experiments in a Syrian hamster model confirmed the high antiviral potency of this drug. Inhalations of animals significantly reduced viral load in lungs (up to 1.5–3-fold) and virus-induced pulmonary inflammation in a dose-dependent manner [126]. 

The results of a randomized controlled open-label clinical trial showed that this drug is safe, well tolerated, and significantly reduces the time to clinical improvement in patients hospitalized with moderate COVID-19 compared to standard therapy based on anti-inflammation and antiviral components approved by Russian healthcare authorities [124]. It was shown that patients from the low-dose group achieved the primary endpoint defined by simultaneous achievement of the relief of fever, normalization of respiratory rate, reduction of coughing, and oxygen saturation of >95% for 48 h significantly earlier (median 6 days; 95% confidence interval [CI]: 5–7, HR 1.75, *p* = 0.0005) than patients from the control group (8 days; 95% CI: 7–10). At the same time, no beneficial effect was found for the high-dose group, which has been shown to be associated with the formation of large molecular aggregates that enter the cell much less efficiently.

The advantage of the siRNA-based approach is that the molecules act directly on the viral genome, thereby reducing the risk of developing resistance to therapy. By contrast, protease inhibitors such as Paxlovid act at the protein level, which opens up additional opportunities for the virus to escape. RNA-viruses have a number of mechanisms for posttranscriptional modification of their protein structure. These include, for example, the use of alternative reading frames and the use of non-canonical start codons [127] which provide opportunities for accelerated evolution of viral proteins and thus, the development of resistance [128]. Moreover, siRNA provides strictly specific action on the virus, since the sequence of the target gene has no homology to human genes, while at the same time it is metabolized quite effectively, which explains its high safety profile and absence of side effects, unlike other drugs in which side effects are often reported [129].

## 5. Conclusions

It is clear that COVID-19 has been and continues to be a serious problem for global health. Sequencing of the viral genome and X-ray crystallographic analysis of its proteins made it possible to describe the life cycle of the pathogen in detail. Current knowledge about the molecular mechanisms of virus entry, genome replication, mRNA transcription, and viral protein translation allows the successful development of new COVID-19 pharmacotherapies. In this review, we summarize mechanisms and emerging clinical data of the pharmacotherapy of COVID-19. Current etiotropic therapy of COVID-19 includes the use of plasma of convalescents, neutralizing monoclonal antibodies, interferons, nucleoside analogs, protease, and fusion inhibitors.

One of the first drugs used for COVID-19 treatment was plasma of the convalescent, containing antibodies against the virus. Initial clinical trials showed promising results, but further multicenter clinical trial and systematic review revealed no clinical efficacy of plasma, which could be due to the fact that plasma contains antibodies not only against the RBD domain, but also against other less significant epitopes of the virus. Despite this, beneficial effect of plasma can be achieved in immunocompromised patients. As an alternative, hyperimmune globulin of convalescent plasma and SARS-CoV-2-specific monoclonal antibodies can be used, but more clinical data supporting the efficacy of these approaches are needed.

Interferons being a first line of antiviral defense of the host were also used for COVID-19 therapy. In general, accumulated clinical experience demonstrates no beneficial effects of interferons for the treatment of the disease because SARS-CoV-2 is able to escape the antiviral effects of interferons. However, certain subgroups of COVID-19 patients may respond to such therapy.

Nucleoside analogs (favipiravir, ribavirin, remdesivir, and molnupiravir) were initially developed against different viruses, while they were repurposed for COVID-19 therapy. Nucleoside analogs inhibit SARS-CoV-2 genome replication resulting in elimination of the pathogen. Ribavirin is currently not used for the treatment of COVID-19 due to the limited efficacy and unacceptable safety profile. At the same time, efficacy of other nucleoside analogs is well defined and therefore, they were approved for clinical use in many countries.

Protease inhibitors targeting high conservative M-pro enzyme, which is extremely important for the SARS-CoV-2 life cycle, represent another class of antivirals. Two drugs, lopinavir and nirmatrelvir, in combination with their pharmacokinetic booster, -ritonavir, are used for COVID-19 treatment. Current clinical data showed no efficacy of lopinavir/ritonavir in the treatment of COVID-19 patients, but it could be beneficial when it is used in combination with other therapies (arbidol, interferons, ribavirin). At the same time, nirmatrelvir/ritonavir (Paxlovid) demonstrates clinical benefits.

Umifenovir (arbidol) is a drug exhibiting antiviral activity through the inhibition of SARS-CoV-2 entry to target cells and stimulation of immune response. It is approved for COVID-19 therapy in Russia and China.

Drugs based on antisense oligonucleotides are a fundamentally new direction in the therapy of viral diseases and primarily SARS-CoV-2. The advantage of such drugs is their high specificity and safety.

Analysis of laboratory and clinical data suggests that current antivirals are able to successfully combat broad spectra of emerging SARS-CoV-2 strains, providing reliable protection against COVID-19.

## Figures and Tables

**Figure 1 ijms-24-08867-f001:**
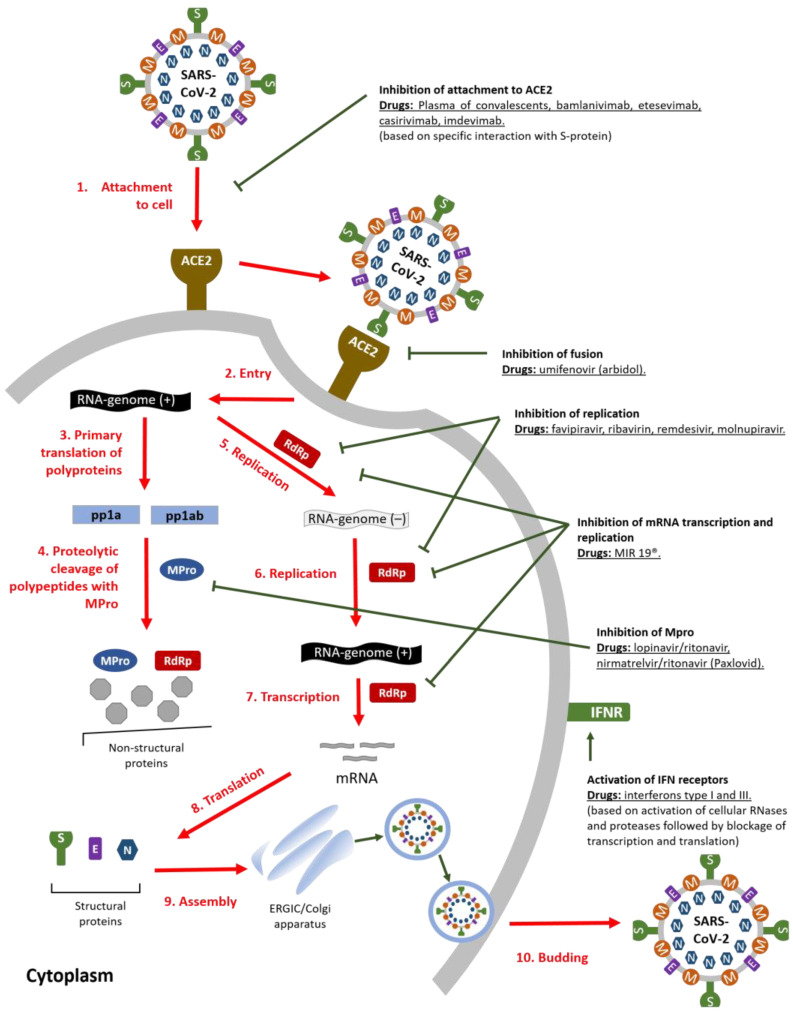
SARS-CoV-2 life cycle and mechanisms of antiviral drugs action.

**Figure 2 ijms-24-08867-f002:**
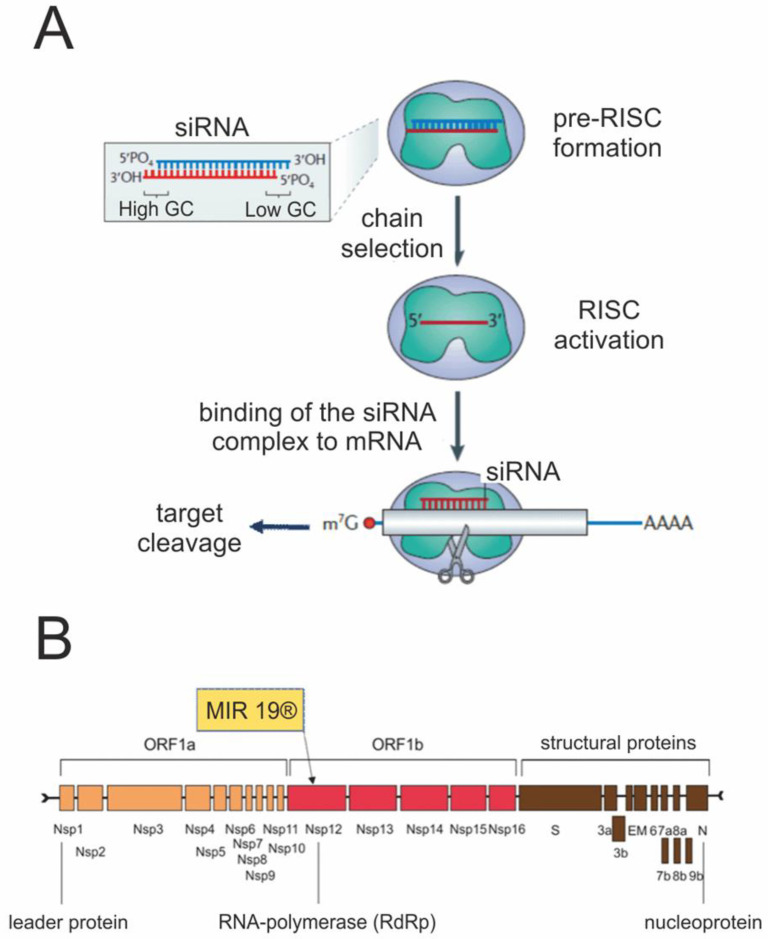
Mechanism of MIR 19^®^ action (**A**) and its biological target (**B**).

**Table 1 ijms-24-08867-t001:** Registered and experimental drugs against SARS-CoV-2.

Type of Therapy	Drug	Mechanism	Recommendations	References
Plasma of convalescents	Plasma of convalescents	Virus binding by specific polyclonal antibodies	Not recommended	[22,23]
Monoclonal antibodies	Bamlanivimab + Etesevimab Casirivimab + Imdevimab Sotrovimab Regdanvimab	Virus binding by S-protein-specific monoclonal antibodies	Recommended	[22,23,24]
	Tocilizumab Sarilumab	Suppression of the SARS-CoV-2 infection induced inflammation by IL-6 receptor inhibition	Recommended	[22,23,25,26,27]
Interferons	Interferon-β	Activate genes encoding antiviral proteins which limit viral invasion, restrict replication of viral genome and viral protein translation. Regulate antigen presentation, function of natural killer cells, and activate B- and T-cells contributing to the viral clearance	Not recommended	[22,23]
Nucleoside analogs	Favipiravir	Inhibition of viral replication	Recommended	[22,23,28]
	Ribavirin		Not recommended	[23,29]
	Remdesivir		Recommended	[23,29]
	Molnupiravir		Recommended	[22,23]
Small molecule- based therapy	Lopinavir/ritonavir	Covalently or non-covalently bound to catalytically important amino acids of viral M-pro enzyme resulting in its functional inactivation.	Not recommended	[30,31,32]
	Nirmatrelvir/ritonavir (Paxlovid)		Recommended	[33,34]
	PLpro inhibitors	Direct inactivation of viral PLpro enzyme.	Experimental therapy	[35]
	Umifenovir	Inhibition of viral and cell fusion	Recommended in China and Russian Federation	[36]
Antisense oligonucleotides	MIR 19^®^	The antiviral effect is based on the suppression of the RdRp gene of the SARS-CoV-2 by the mechanism of RNA interference	Recommended in Russian Federation	[36]

## Data Availability

Not applicable.

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
