# Peer review of "Antiviral Therapy of COVID-19"

_ijms, 2023, doi:10.3390/ijms24108867_

Round 1
Reviewer 1 Report (New Reviewer)
This comprehensive review of anti-COVID-19 therapeutic drugs or antivirals is sound except that minor English revision is needed. For example,
1. Abstract: Change the sentences "The declining... pathogen, has shifted the focus... life cycle was published" (lines 12-15) to "Due to declining... pathogen, control of the disease has shifted to the focus... life cycle has been published". Replace the phrase "recommendations of COVID-19" (line 18) with "recommendations for COVID-19".
2. Introduction (paragraph 1): Change the phrase "continues spreading" (line 27) to "continues posing threat to public health". Insert the missing conjunction "and" before the phrase "more than" (line 28). Change the phrase "to develop treatment algorithms" (lines 32 to 33) to "to improve treatment efficacy".
3. check the accuracy of the sentence "The first two genera mainly infect mammals, whereas the gammacoronaviruses infect birds, and the latter infect both" (lines 47 to 48).
4. Delete the redundant word "spread" (line 250).
5. Change the word "repositioning" (lines 328 and 475) to "repurposing", and the word "repositioned" (line 545) to "repurposed".
6. Change the first "MIR 19" (line 483) to "MIR 19® (siR-7-EM/KK-46)", and the remaining "MIR 19" (line 570 and last row of Table 1) to "MIR 19®".
Author Response
Please see the attachment

Reviewer 2 Report (New Reviewer)
This review presented the status of anti-COVID-19 therapies which have been through clinical trials worldwide. The summary is informative and covering broad range of drugs and their mechanism of action. However, the sectional classification is somewhat indistinct as it includes both types of drugs and targets. While most sectional titles refer to types of drug, sections 3.4 and 3.6 are describing small molecules targeting different proteases. Because of the popularity of the main target, COVID-19, it could mislead readers who do not have a deep understanding of this field. Also, section 3.5 is missing. Therefore, it is suggested to combine sections 3.4 and 3.6 under the title of small molecule-based therapy. Table 1 needs to be revised accordingly.
Lines 166-168, A reference of imatinib for COVID-19 induced inflammatory treatment is missing.
Line 142. What is the standard treatment?
Author Response
Please see the attachment

This manuscript is a resubmission of an earlier submission. The following is a list of the peer review reports and author responses from that submission.
Round 1
Reviewer 1 Report
In the review, the authors summarize different strategies aiming to combat COVID-19 disease, including spike monoclonal antibodies, interferons, viral RdRp inhibitors, viral main protease inhibitors, and RNAi oligoes targeting the RdRp. Overall, this is a well-written review and will be of interest to the cell biologists in the SARS-CoV-2 field. However, there are some obvious mistakes that must be corrected and improvements in writing and reference should be done before acceptance to publish in the International Journal of Molecular Sciences.
Major points:
1. In Figure 1, the SARS-CoV-2 viral particle contains three structural proteins, S, E, and N. Where is M protein?
2. Also in Figure 1, viral particles are supposed to be assembled in the ERGIC/Golgi compartment, and viruses transport through the Golgi apparatus and are released through membranous vesicles, which should be included in the model.
3. In section 2.5, the virions assembly and budding part need to be improved. Proteins S, M, and E are synthesized in the ER and assembled in the ERGIC/Golgi. It is reported that SARS-CoV-2 downregulates Golgi structural proteins to induce Golgi fragmentation to facilitate trafficking. So, newly assembled virions must traffic through the Golgi apparatus before being released to the extracellular space.
4. In line 70, SARS-CoV-2 encodes at least 14 ORFs, but not 5. The No 11 reference is not suitable here and the authors should find a better one. Similarly, in line 104, there are at least 8 ORFs encoding vial accessory proteins, but not 5. There are ORF3a, 3b, 6, 7a, 7b, 8, and 10. In Figure 2, the authors listed 8 ORFs for accessory proteins, ORF8a and ORF8b are not well documented compared to ORF10. Anyway, there are 8 ORFs for accessory proteins, 5 is definitely not right.
5. NSP5 (Main protease, 3C-like protease) inhibitors are extensively discussed, but the NSP3 (Papain-like protease) inhibitors are also widely studied and should be discussed.
Minor points
1. There are a lot of excellent structural studies showing the binding of the inhibitors to the viral proteins, like M protease, which will be valuable for readers if discussing a few examples.
2. Some spelling mistakes should be modified. For example, in line 200, result in should be results in; in line 237, Authors conclude that INF-b is being a safe agent does not appear…, add “but” after agent or modified in another way; in line 401, prolong should be prolongs; in line 479, allows successful development or modified; in line 494, because SARS-CoV-2 able to escape, add is before able.
3. In line 451, siRNA (usually 21-26 bp in size), some paper claims usually 20-25 bp, please clarify.
Author Response
Thank you for working with and reviewing our article. The text has been corrected according to your comments. Please find point-by-point response in attachment

Reviewer 2 Report
section 2 :SARS-CoV-2 life cycle:
the section is very long (20% of the writen manuscript)
unneccessary duplications :(sentences start at lines 61 and 83)
line 49- the clinical description of the disease is partial.
section 3:etiotropic therapy COVID-19. official recommendations and medications
1. in this sections the authors review medical therapies for COVID. there is no mention or reference to official guidelines /recommendations of agencies , international organizations or proffesional societies like CDC WHO and others.
2. for some therapeutic interventions the official recent recommendations are against their use : interferon; lopinavir/ritonavir. it is expected from the authors to refer to these recommendations
3. other therapeutic options like Ribavirin are not in clinical use in the western oriented countries.
4. line 161 and on: the authors mention a drug combined of globulines and components that alleviate the condition of patients. it is not clear what the meaning of this is.
5. Umifenovir: the authors' refference for this compund is a clinical trial of very low quality and not suitable for any recommendation ( the trial described in refference 103)
6. antisense oligonucleotides: for this novel therapy the authors refer to 2 clinical trials (references no. 110, 112) results of these trials had not been published .
references:
some of the references can not be accessed (references no. 25, 89)
Author Response

(The authors gave the same response as above.)

Round 2
Reviewer 1 Report
In the revision, the authors addressed most questions. The manuscript was much improved, which can be considered to be accepted for publication after minor revision. Since Golgi fragmentation is one of the most dramatic morphological alterations after SARS-CoV-2 infection, it should be included in the assembly/trafficking part (section 2.5). These papers (https://www.sciencedirect.com/science/article/pii/S193131282030620X; https://www.mdpi.com/1999-4915/13/9/1798; https://www.biorxiv.org/content/10.1101/2022.03.04.483074v2) experimentally reported the clear observation and potential mechanisms of Golgi fragmentation induced by SARS-CoV-2, which should be cited.
Author Response
Point 1: In the revision, the authors addressed most questions. The manuscript was much improved, which can be considered to be accepted for publication after minor revision. Since Golgi fragmentation is one of the most dramatic morphological alterations after SARS-CoV-2 infection, it should be included in the assembly/trafficking part (section 2.5). These papers (https://www.sciencedirect.com/science/article/pii/S193131282030620X; https://www.mdpi.com/1999-4915/13/9/1798; https://www.biorxiv.org/content/10.1101/2022.03.04.483074v2) experimentally reported the clear observation and potential mechanisms of Golgi fragmentation induced by SARS-CoV-2, which should be cited.
Response 1: Thank you for your comment. We have supplemented the text of the article in the appropriate section and provided a link to the article by Cortese et al. https://www.sciencedirect.com/science/article/pii/S193131282030620X
Reviewer 2 Report
1. you should consider divide all therapeutic interventions to recoommended, not currently recommended and experimental.
2. for the experimental agents
PL-pro : it has to be mentioned that the discussion is theoretical as there are no mentioned clinical trials in their regard
MIR-19 - you refer to 1 clinical study. this study is very small , is open label and statistical significant results of one parameter was observed only for the low dose. this necessitates explanation and more importantly more data.
so you have to rephrase accordingly the abstract and conclusions sections
you mentioned that this agent is approved? by which health authority? can you provide refference?